# Ribosome Biogenesis Serves as a Therapeutic Target for Treating Endometriosis and the Associated Complications

**DOI:** 10.3390/biomedicines10010185

**Published:** 2022-01-17

**Authors:** Cherry Yin-Yi Chang, An-Jen Chiang, Man-Ju Yan, Ming-Tsung Lai, Yun-Yi Su, Hsin-Yi Huang, Chan-Yu Chang, Ya-Hui Li, Pei-Fen Li, Chih-Mei Chen, Tritium Hwang, Chloe Hogg, Erin Greaves, Jim Jinn-Chyuan Sheu

**Affiliations:** 1Department of Obstetrics and Gynecology, China Medical University Hospital, Taichung 404332, Taiwan; d4754@mail.cmuh.org.tw; 2Department of Medicine, School of Medicine, China Medical University, Taichung 404333, Taiwan; 3Department of Obstetrics and Gynecology, Kaohsiung Veterans General Hospital, Kaohsiung 813414, Taiwan; ajchiang490111@gmail.com; 4Institute of Biomedical Sciences, National Sun Yat-sen University, Kaohsiung 804201, Taiwan; may781111@gmail.com (M.-J.Y.); yunyisue@gmail.com (Y.-Y.S.); angel86916@gmail.com (H.-Y.H.); peter80168@gmail.com (C.-Y.C.); sophia19960214@gmail.com (Y.-H.L.); acd6036@gmail.com (P.-F.L.); tritium@mail.nsysu.edu.tw (T.H.); 5Department of Pathology, Taichung Hospital, Ministry of Health and Welfare, Taichung 403301, Taiwan; lukemtlai@gmail.com; 6Human Genetic Center, China Medical University Hospital, Taichung 404332, Taiwan; t6783@mail.cmuh.org.tw; 7Medical Research Council Centre for Reproductive Health, The University of Edinburgh, Edinburgh EH16 4TJ, UK; chloe.x.hogg@gsk.com; 8Centre for Early Life, Warwick Medical School, University of Warwick, Coventry CV4 7AL, UK; 9Department of Biotechnology, Kaohsiung Medical University, Kaohsiung 807378, Taiwan; 10Institute of Biopharmaceutical Sciences, National Sun Yat-sen University, Kaohsiung 804201, Taiwan; 11Institute of Precision Medicine, National Sun Yat-sen University, Kaohsiung 804201, Taiwan

**Keywords:** endometriosis, ribosome biogenesis, RNA polymerase 1, c-MYC/mTOR signaling, inflammation, pain

## Abstract

Ribosome biogenesis is a cellular process critical for protein homeostasis during cell growth and multiplication. Our previous study confirmed up-regulation of ribosome biogenesis during endometriosis progression and malignant transition, thus anti-ribosome biogenesis may be effective for treating endometriosis and the associated complications. A mouse model with human endometriosis features was established and treated with three different drugs that can block ribosome biogenesis, including inhibitors against mTOR/PI3K (GSK2126458) and RNA polymerase I (CX5461 and BMH21). The average lesion numbers and disease frequencies were significantly reduced in treated mice as compared to controls treated with vehicle. Flow cytometry analyses confirmed the reduction of small peritoneal macrophage and neutrophil populations with increased large versus small macrophage ratios, suggesting inflammation suppression by drug treatments. Lesions in treated mice also showed lower nerve fiber density which can support the finding of pain-relief by behavioral studies. Our study therefore suggested ribosome biogenesis as a potential therapeutic target for treating endometriosis.

## 1. Introduction

Endometriosis is a complex, hormone dependent inflammatory disorder that affects more than 10% of women of reproductive age [1,2]. This enigmatic disorder is characterized by the presence of multi-cellular ectopic endometrial tissue deposits (lesions) within the peritoneal cavity that undergo proliferation, neuroangiogenesis and immune cell infiltration. Women with endometriosis usually suffer from chronic pelvic pain, dysmenorrhea, and infertility. Although several theories have been proposed regarding the etiology of endometriosis, the exact pathogenesis of the disease and possible treatment strategies require further investigation [1,2]. Surgery, pain medication or ovarian suppression are common approaches for managing endometriosis. However, the recurrence rate after surgery is high, and current hormonal treatments show no therapeutic efficacy and have unwanted side-effects. Thus, there is an unmet clinical need for effective therapies that can suppress endometriosis growth and the associated complications.

At molecular levels, the survival and growth of ectopic endometrial cells on peritoneal surfaces and invasion into visceral tissue can be likened to tumorigenesis, in so far as ectopic endometrial tissue is accompanied by angiogenesis and cell migration/invasion [3,4]. Histopathological observations and genetic analyses also provide evidence that endometrioid and clear-cell types of ovarian carcinoma can arise from endometriotic lesions [5,6]. Notably, sustained up-regulation of global protein synthesis is the most unique characteristic for cells to gain metabolic advantages in cell proliferation [7,8], survival and malignant transformation [9,10,11]. In our previous study, a concerted overall increase of small nucleolar RNAs (snoRNAs) and ribosomal proteins (RPs) was found in endometriotic lesions collected from patients, and such phenomenon was even more pronounced in associated ovarian cancer [12]. Both snoRNAs and RPs have been known as key regulators in ribosome biogenesis that tightly regulate cell-cycle progression, cellular senescence/death, nuclear architecture and global gene expression. Therefore, it is highly possible that activation of ribosome biogenesis can serve as a driving force for unfavorable cell survival and invasion during endometriosis progression.

Development of new effective therapies largely depends on access to good animal models that offer physiologically relevant features of human disorders. To consider lower cost and smaller animal size, models using mice with an intact immune system are a suitable option. Currently, most studies have involved the introduction of small fragments of mouse uterine tissue sutured to the peritoneum or other sites [13,14]. However, mice do not exhibit spontaneous decidualization or menstruation, thus previous approaches do not recapitulate the tissue microenvironment at the time of retrograde menstruation. To overcome this limitation, we used a mouse model of induced endometriosis that phenocopies many features of endometriosis in women by introducing syngeneic ‘menses’-like endometrium into the peritoneal cavity of immunocompetent mice [13,15]. The model displays robust changes in sensory behavior indicative of endometriosis-associated hyperalgesia and associated molecular alterations in the nervous system [16].

Ribosome biogenesis is one of the most multifaceted and energy-demanding processes in cells, involving multiple cellular inputs such as mitogenic signals and nutrient availability. Perturbation of ribosome assembly by inhibiting RNA polymerase I or by silencing snoRNAs/RPs has been suggested as a novel strategy against malignant diseases which can arrest cancer cell proliferation and induce apoptosis through the MDM2-p53-p21 axis [17,18,19]. Due to the lack of effective treatments, those findings provide us the rationale to evaluate the feasibility of anti-ribosome biogenesis therapy against endometriosis and the associated complications by using a preclinical mouse model. Moreover, other treatments active in cancer therapy have proven to be promising as potential treatments for endometriosis [20]. To this end, we treated endometriosis mice with mTOR/PI3K inhibitor (GSK2126458) and RNA polymerase I inhibitors (CX5461 and BMH21) to shut down the upstream signaling for protein biosynthesis and subsequent rDNA transcription, respectively. The therapeutic efficacy was further validated by flow cytometry analysis to evaluate the impact on peritoneal immune milieu, sensory behavior and tissue staining to investigate neuro-angiogenesis. Results from this study can provide molecular evidence to address whether activation of ribosome biogenesis functions as a driving force to promote endometriosis pathogenesis and whether it could be utilized as a newly-defined target for treating this debilitating disorder. 

## 2. Materials and Methods

### 2.1. Mouse and Reagents

Wild-type C57BL/6JNarl female mice were purchased from National Laboratory Animal Center (NLAC, Taipei, Taiwan) at 8 to 12 weeks of age. Anti-ribosome biogenesis agents were purchased from AdooQ BioScience (Irvine, CA, USA), including the mTOR/PI3K inhibitor: GSK2126458 (A11035) and RNA polymerase-I inhibitors: CX5461 (A11065) and BMH-21 (A14335).

### 2.2. Mouse Model of Induced Endometriosis

The mouse model of induced endometriosis was established by using syngeneic menstrual endometrial tissue introduced into the peritoneum of immunocompetent mice (Figure 1a). The detailed protocol for this mouse model was described in the previous study [15]. Based on this model, the mice can develop not only endometriotic lesions but also several phenotypes that can be detected in human bodies, including inflammatory mediators, dynamics of tissue-resident immune cells and neuron fiber formation associated with the lesions. Power analyses based on our pilot data suggest *n* = 20–25 animals per treatment that provides 80% power to detect 5% significance in this study. The conditions for drug treatments were designed based on protocols utilized in the previous studies [21,22], and they were 75 μg in 100 μL/mouse/day for GSK2126458 (mTOR/PI3K inhibitor) and 1.25 mg in 100 μL/mouse/day for RNA polymerase-I inhibitors including CX5461 and BMH-21. The drugs were administered orally five days per week for three weeks (Figure 1a).

### 2.3. Immune Cell Profiling by Flow Cytometry Analysis

After sacrifice, peritoneal lavages were collected from mice by injecting 5 mL serum-free DMEM into the peritoneal cavity. Red blood cells inside the samples were lysed and the rest cells were then stained with a combination of antibodies shown in Appendix A. Just prior to analysis, 123count eBeads (Thermo Fisher Scientific, Waltham, MA, USA) were added into the samples, allowing absolute numbers of cells to be determined. Flow cytometry was conducted to analyze immune cell population by using LSR II Flow Cytometer (BD Biosciences, Franklin Lakes, NJ, USA). The gating strategies for neutrophils and macrophages were shown in Figure 2a and Figure 3a, respectively. The data was analyzed with FlowJo v.10 software (FlowJo LLC., Ashland, OR, USA).

### 2.4. Behavior Assessments

Behavior assessments were performed as described in the previous study [16], including spontaneous (grooming and activity) and evoked (mechanical hyperalgesia measured using von Frey filaments) behaviors. During the assessment, the behaviors were recorded in a blinded fashion, and all animals were acclimatized to the apparatus and handling prior to the beginning of behavior analysis.

### 2.5. Sandwich ELISA

The supernatants of peritoneal lavages were collected by centrifugation and subjected to sandwich ELISA study. The ELISA kits were purchased from Thermo Fisher Scientific (EM9RB for mouse β-NGF and EMIGF1 for mouse IGF-1, Waltham, MA, USA). The experiments were performed according to the protocols provided by the company. Absorbance of colorimetric signal was measured at 650 nm with iMark microplate reader (Bio-Rad Lab., Hercules, CA, USA). The absolute concentration was estimated by the standard curve using the recombinant protein from the kit.

### 2.6. Real-Time qPCR

RNA samples were extracted from peritoneal tissue, spinal cord and posterior insula in the brain of experimental mice by using RNA extraction kit (FairBiotech, Taoyuan, Taiwan) and subjected to cDNA synthesis immediately by using High-Capacity cDNA Reverse Transcription Kit (Thermo Fisher Scientific, Waltham, MA, USA). Real-time qPCR was carried out in triplicate to estimate expression levels of inflammatory cytokines with primers listed in Appendix A. The cycling condition for PCR reactions includes initial denaturation at 95 °C for 1 min followed by 45 cycles of 95 °C for 15 s, 60 °C for 15 s, and 68 °C for 15 s. GAPDH levels were utilized as the internal controls for data normalization.

### 2.7. Immunohistochemistry (IHC) and Immunofluorescent (IF) Staining

Tissue sections of endometriotic lesions were prepared for IHC and IF staining with antibodies listed in Appendix A. The detailed procedures can be found in the previous studies [15,23]. The images were taken by an IX83 fluorescent microscope (Olympus Corp., Shinjuku, Japan) and the fluorescence staining intensity was analyzed by Image-Pro Premier v.10 (Media Cybernetics Inc., Rockville, MD, USA).

### 2.8. Statisitc Analysis

GraphPad Prism (GraphPad software v.5, La Jolla, CA, USA) and SPSS v.14.0 software (SPSS Inc. Chicago, IL, USA) were utilized for statistical analyses in this study. Student’s *t*-test was performed to show statistical differences between two groups. One-way ANOVA was used to compare differences among multiple groups. The data were presented as mean ± S.D., and a *p* value less than 0.05 was considered as significant difference.

## 3. Results

### 3.1. Blockage of Ribosome Biogenesis Led to Reduced Formation of Endometriotic Lesions

Using syngeneic ‘menses-like’ endometrial tissue introduced into the peritoneal cavity of immunocompetent mice, we have previously demonstrated the establishment of endometriotic lesions in abdomen of the recipient mice [15]. During endometriosis development, the mTOR/PI3K inhibitor GSK2126458 (75 μg in 100 μL/mouse/day) and RNA polymerase-I inhibitors, CX5461 and BMH-21 (1.25 mg in 100 μL/mouse/day), were administered orally five days per week for three weeks (Figure 1a). The resultant lesions that contain stromal (anti-vimentin) and/or epithelial (anti-cytokeratin) cell compartments were collected and counted (Figure 1b). A well-developed vasculature and nerve fibers were also found inside the lesions with macrophage infiltration (Figure 1c). In this study, the incidence rate of endometriosis in the control group treated with vehicle only was 80% with an average lesion number of 1.80 per mouse. As compared to the control, mice treated with GSK2126458 and CX5461 exhibited significantly decreased endometriosis incidence rate (GSK2126458: 52.2%; CX5461: 40.0%) (Figure 1d, upper panel). The average lesion numbers were also significantly reduced in those mice that did develop endometriosis (GSK2126458: 0.78; CX5461: 0.72; BMH-21: 0.88) (Figure 1d, lower). Our data suggests that ribosome biogenesis can be utilized as a therapeutic target for treating endometriosis.

### 3.2. Anti-Ribosome Biogenesis Modifies the Peritoneal Immune Milieu

The peritoneal immune environment is central to the pathophysiology of endometriosis. To evaluate the impact of anti-ribosome biogenesis agents on the peritoneal immune milieu, peritoneal fluids were collected from control and treated mice on day 40 (Figure 1a) and subjected to flow cytometry analyses. Neutrophils in the peritoneal fluid was measured by using the gating strategy shown in Figure 2a. Our data revealed that mice with induced endometriosis showed much higher neutrophil numbers in the peritoneal cavity as compared to the levels in naïve mice (*p* value < 0.0001) (Figure 2b,c). Notably, mice with induced endometriosis treated with GSK2126458 and CX5461 showed significant reduction of neutrophil levels (*p* values were 0.0047 and 0.0002, respectively) (Figure 2c), suggesting suppression of chronic inflammation in their peritoneal cavity by those inhibitors. BMH-21 showed partial suppressing effects but did not reach statistical significance.

We also analyzed the dynamic profile of small (SpM) and large (LpM) peritoneal macrophages (Figure 3a), which play important roles in modulating immune microenvironment in the peritoneal cavity [24,25]. As shown in Figure 3b,c, drug treatments significantly maintained SpM population at low levels, similar to the levels in the peritoneal cavity of naive mice. BMH-21 was able to reduce numbers of LpM in mice with endometriotic lesions (*p* = 0.0143). The SpM/LpM ratios were significantly reduced in mice treated with anti-ribosome biogenesis agents (*p* values were GSK2126458: 0.0024, CX5461: 0.0013, and BMH-21: 0.0019). Collectively, these data combined with the neutrophil analysis in Figure 2 suggest that anti-ribosome biogenesis can reduce the inflammatory milieu in mice with induced endometriosis.

### 3.3. Anti-Ribosome Biogenesis Agents Suppressed Concentrations of Peritoneal Neurotrophic Factors and Nerve Fiber Growth in Endometriotic Lesions

It is well established that endometriotic lesions become infiltrated by nerve fibers, and this is thought to contribute to endometriosis-associated hyperalgesia. Our previous studies revealed a role for macrophages in promoting nerve fiber growth and associated pain in mice with induced endometriosis [23,26,27]. Thus, we aimed to investigate whether the suppression of the inflammatory milieu by anti-ribosome biogenesis may also provide additional benefits in managing pain. NGF and IGF-1, key neurotropic factors known to contribute to nerve fiber growth and activation, were analyzed by sandwich-ELISA, and we found that mice with induced endometriosis treated with GSK2126458 or CX5461 exhibited much lower concentrations of NGF and IGF-1 in the peritoneal cavity compared to vehicle-treated mice (Figure 4a; *p* values for NGF, GSK2126458: 0.0162, CX5461: 0.0489, naïve: 0.0061; *p* values for IGF-1, GSK2126458: 0.0228, CX5461: 0.0136, naïve: 0.0161). To confirm the effectiveness of the inhibitors in regulating nerve fiber growth, tissue sections of endometriotic lesions were prepared for PGP9.5 staining, a neuron specific protein biomarker. Consistent with the ELISA data, less neuronal cells were detected in lesions from mice treated with GSK2126458 or CX5461 as compared to lesions collected from vehicle-treated mice (Figure 4b; *p* values were 0.0002 and 0.0003, respectively). Lesions from BMH21-treated mice also showed less PGP9.5 staining but did not reach statistical significance (*p* = 0.0777).

### 3.4. Anti-Ribosome Biogenesis Suppressed Chronic Inflammation-Induced Hyperalgesia

Chronic endometriosis-associated hyperalgesia is associated with long-term peripheral inflammatory stimuli within the peritoneal cavity, resulting in maladaptation within the nervous system that amplifies the pain response [28,29]. To test the effectiveness of anti-ribosome biogenesis inhibitors on inflammatory mediators in the peritoneal tissue and central nervous system, we measured mRNA levels of four key inflammatory cytokines in peritoneum, spinal cord and posterior insula in the brain by quantitative RT-PCR (Figure 5). Our data indicated that mRNA levels of COX-2, IL-1β and TNF-α were significantly enhanced in peritoneum during the development of endometriotic lesions. Up-regulation was also detected in the spinal cord of mice with induced endometriosis, whilst TNF-α was the only upregulated cytokine found in the posterior insula (Figure 5d). Anti-ribosome biogenesis reagents partially suppressed mRNA expression of COX-2 and TNF-α in the spinal cord and brain but not in peritoneal tissue (Figure 5a,d). GSK2126458 or CX5461 suppressed mRNA levels of IL-1β (Figure 5c) in both peritoneal tissue and spinal cord. IL-6 levels fluctuated in different tissues (Figure 5b), probably due to its pleiotropic and complex roles in inflammation [30].

Next, we aimed to determine the impact of anti-ribosome biogenesis agents on sensory behavior of mice with induced endometriosis. Firstly, we investigated loco-motor activity of treated mice running through an enrichment tunnel. As shown in Figure 6a, the tunnel-entering activity of vehicle-treated mice was 7.6 ± 3.2 times in 5 minutes, which is much lower than 18.3 ± 2.9 times in naïve control group, suggesting chronic pain-induced decrease in activity. When compared to vehicle group, mice treated with anti-ribosome biogenesis reagents showed higher activity levels. Observations on abdominal grooming behaviors revealed that CX5461 could reduce abdominal grooming in mice with induced endometriosis, compared to the vehicle-treated mice (Figure 6b). We observed that in some mice GSK2126458 could induce skin allergies, resulting in treated mice exhibiting higher abdominal grooming frequencies. The von Frey filament test for mechanical hyperalgesia on the abdomen and hind paw were also performed and our data indicated reduced pain sensitivity in mice with induced endometriosis treated with GSK2126458 or CX5461 (Figure 6c,d) as compared to mice treated with vehicle.

## 4. Discussion

Transcription of rRNA gene is a key regulatory step to maintain the availability of growth factors, nutrients and energy in proliferating cells. In this study, our data provided the first evidence to support anti-ribosome biogenesis as a possible therapeutic strategy to treat endometriosis. Figure 7 summarizes our rationale and findings that inhibition in ribosome biogenesis either by blocking upstream PI3K/AKT/mTOR signaling or by direct shutdown of RNA polymerase-1 in a mouse model of endometriosis that presents human phenotypes [15]. Among the potential drugs tested in this study, GSK2126458 or CX5461 showed therapeutic effects on endometriosis growth (Figure 1) and associated inflammation (Figure 2, Figure 3 and Figure 5), resulting in suppression of nerve fiber growth (Figure 4) and pain relief (Figure 6). We noticed BMH-21 showed low solubility in the vehicle which should be a key issue limiting its therapeutic potency in our study.

The cell nucleolus is the main machinery involved in ribosome biogenesis and emerging evidence indicate its novel roles beyond protein production in sensing and responding to endogenous and exogenous stresses, which are coupled to regulation of cell-cycle progression, cellular senescence/death, nuclear architecture and global gene expression. Accumulating evidence implicate the involvement of ribosome biogenesis in controlling cell fate and carcinogenesis through a bypassing of ribosomal/oncogenic stress responses [9,10,11]. Previous study confirmed differential therapeutic effects of CX5461 against cancer depending on the genetic status of *TP53* [22]. Notably, *TP53* mutation is a relatively rare genetic event (around 10%) during endometriosis development and progression [31]. Furthermore, endometriosis-associated malignancies, such as clear cell type or endometrioid type ovarian cancers, also do not show frequent *TP53* mutation event, in contrast to serous type ovarian cancer which can sum up to more than 80% cases [31,32,33]. Activation of p53-mediated cell death signaling therefore could be one of possible mechanisms triggered by anti-ribosome biogenesis agents, resulting in less lesions in treated mice.

The impact of anti-ribosome biogenesis on chronic inflammation and the associated pain were significant based on our mouse study. Currently, limited knowledge is understood for the casual relationship between nucleolar hyperactivation/stress and immune regulation. Recent study revealed that certain cytokines including IL-1β, IL-17A, and TNF-α caused by long-term inflammation can induce aberrant mTOR activity which led to enhanced cell proliferation [34]. IL-6 was also previously reported to upregulate rRNA transcription and ribosome biogenesis through stimulation of c-Myc mRNA translation [35,36]. Genetic study in animal models confirmed that dysregulation of ribosome biogenesis can trigger constant inflammation with expression of pro-inflammatory cytokines in abdomen and infiltrations of macrophages and neutrophils [37,38]. On the other hand, studies on host-pathogen interaction and cellular stress responses revealed co-occurrence of genes involved in both inflammation and ribosome biogenesis [39,40,41]. The above findings indicate a complex network between inflammation and ribosome biogenesis in regulating the microenvironment during endometriosis development. Although multifaceted, PI3K/AKT/mTOR-mediated signaling was considered as a key player to link inflammation with protein synthesis and cell proliferation [42,43,44,45].

The PI3K/AKT/mTOR signaling functions as the upstream regulator of ribosome biogenesis, thus plays a key role in protein synthesis. Transcriptome and protein expression analyses indicated higher levels of genes involved in PI3K/AKT/mTOR signaling in endometriotic lesions than in normal endometria [46,47], especially for patients at advanced stages (stage III to IV) [48] or patients with postmenopausal endometriosis [49]. Functional studies also confirmed increased invasiveness by upregulation of PI3K/AKT/mTOR signaling in endometrial stromal cells, leading to progression of endometriosis [50]. Thus, these findings provided the rationale to treat endometriosis with PI3K/AKT/mTOR inhibitors [51]. By using disease models in mice and rats, a number of studies confirmed the effectiveness of AKT/mTOR inhibitors in reduction of lesion growth [52,53,54] and associated pain [55]. In this study, a PI3K/mTOR dual inhibitor, GSK2126458, was tested in a mouse model of induced endometriosis and proven to be effective in inhibition of lesion growth, chronic inflammation and pain related behaviors. Combination therapy of anti-RNA polymerase-1 agents with PI3K/AKT/mTOR inhibitors could be another promising approach to treat endometriosis and the associated malignancies.

Other epigenetic factors may also contribute to up-regulation of ribosome biogenesis and/or its upstream PI3K/AKT/mTOR signaling, including immune cells, stem cells, adhesion molecules, extracellular matrix metalloproteinases (MMPs), and hormones [56,57,58]. Peritoneal microenvironment altered by those factors during disease progress may play a pivotal role to create the conditions for differentiation, adhesion, proliferation and survival of ectopic endometrial cells. More study might be needed to further investigate the possible links of those epigenetic factors with ribosome biogenesis in promoting the establishment and persistence of endometriotic lesions.

## 5. Conclusions

In sum, our study indicated potent roles of ribosome biogenesis during endometriosis progression. Inhibition of RNA polymerase-1 or PI3K/AKT/mTOR signaling with specific inhibitors significantly suppress lesion growth. Such treatments can also reduce inflammatory conditions in the peritoneal cavity and ease the associated pain in an induced mouse model. Ribosome imbalance by unequal pairing between large and small ribosomal proteins have been previously found to generate a nucleolar stress, leading to p53-mediated apoptosis [59,60]. Gene targeting directly on abnormally upregulated ribosomal proteins (e.g., RPLP2 or PRL38) or rRNA-editing nuclear RNA (e.g., SNORD116) in endometirotic lesions [12] could be an alternative approach which are now under investigation.

## Figures and Tables

**Figure 1 biomedicines-10-00185-f001:**
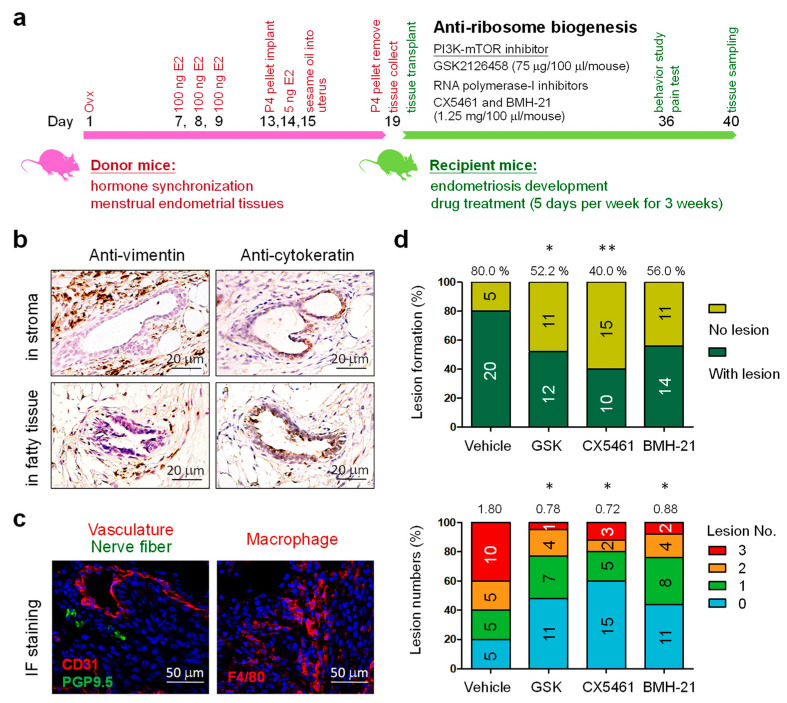
Anti-ribosome biogenesis treatment in a mouse model of endometriosis induced in C57BL/6JNarl mice. (**a**) The schematic diagram indicates the timeline of procedures performed on donor and recipient mice. The recipient mice were further treated with GSK2126458 (GSK; *n* = 23), CX5461 (*n* = 25) and BMH-21 (*n* = 25) at the indicated drug dosages for three weeks. Vehicle-treated mice (*n* = 25) were utilized as controls. (**b**) Tissue section staining revealed the presence of stroma (anti-vimentin) or glands (anti-cytokeratin) in the collected lesions. (**c**) The disease frequencies and average lesion numbers in drug-treated mice were counted and compared with that in control mice. (**d**) Statistical differences between vehicle and drug-treated groups were compared by chi-squire test. The *p* values were presented as *: *p* value < 0.05 and **: *p* value < 0.01.

**Figure 2 biomedicines-10-00185-f002:**
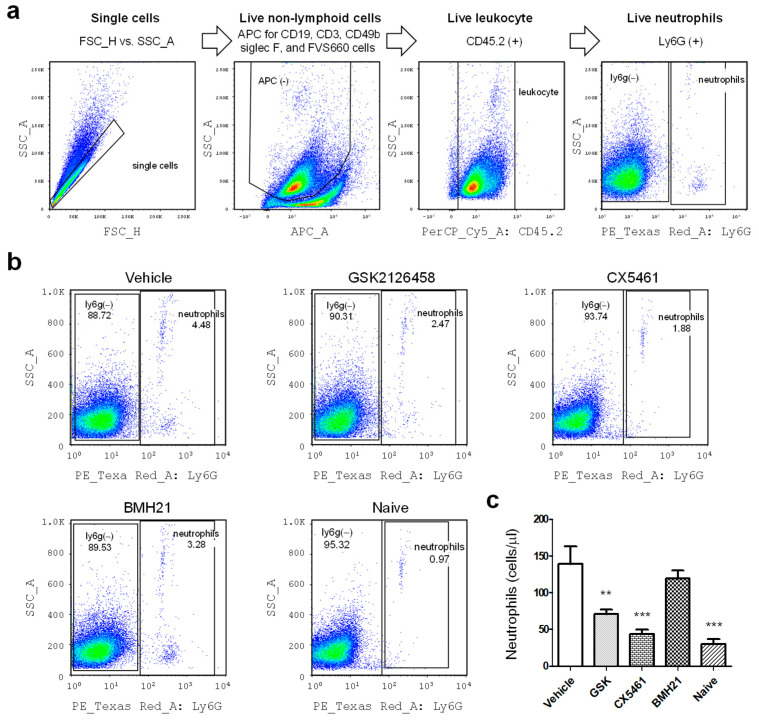
Regulation of inflammation in the peritoneal cavity of mice with induced endome triosis by anti-ribosome biogenesis agents. (**a**) Multicolor flow cytometry was used to assess myeloid cell populations in the peritoneal fluid of mice with induced endometriosis. Neutrophils (Ly6G^+^) were quantified by the sequential gating strategy. (**b**) The depicted cellular gating is representative of five individual experiments (*n* = 25 in each group). The naive group serves as the negative control. (**c**) The bar chart summarized the calculated numbers of neutrophils in each group. Statistical differences between vehicle and drug-treated groups were compared by using *t*-test. The *p* values were presented as **: *p* value < 0.01, and ***: *p* value < 0.001.

**Figure 3 biomedicines-10-00185-f003:**
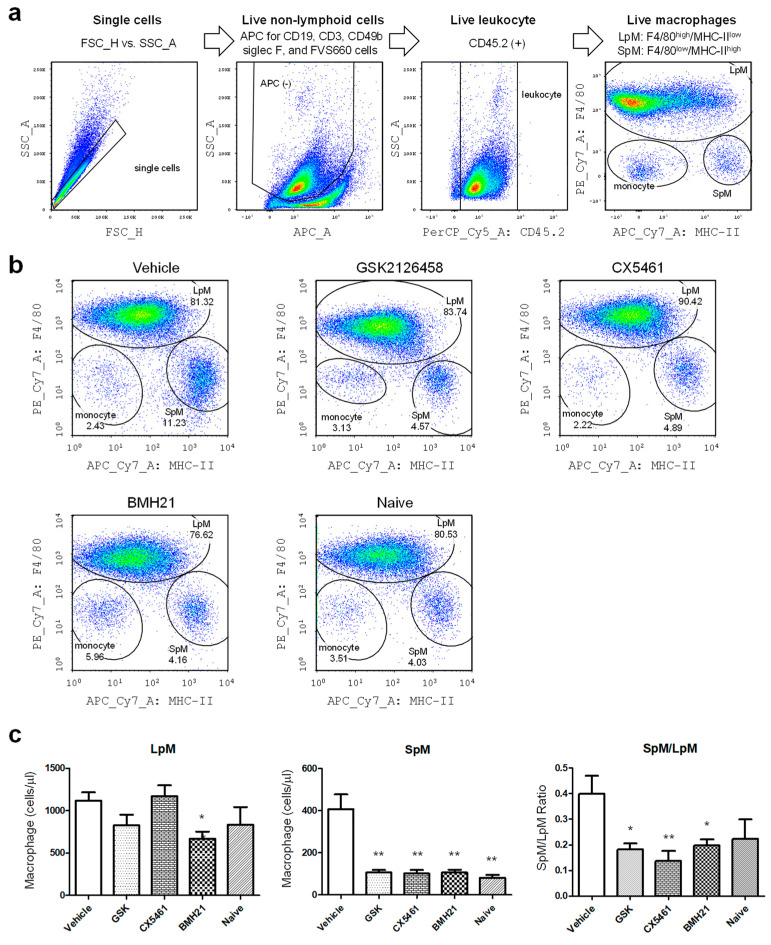
Regulation of macrophage populations in the peritoneal cavity of mice with induced endometriosis by anti-ribosome biogenesis agents. (**a**) Multicolor flow cytometry was applied to gate the myeloid cell populations in the peritoneal fluid of mice with induced endometriosis. Macrophage subsets from each mouse were further analyzed by detecting the expression of F4/80 for large peritoneal macrophages (LpM) or MHC-II for small peritoneal macrophages (SpM). (**b**) The depicted cellular gating is representative of five individual experiments (*n* = 25 in each group). Naïve mice serve as the negative controls. (**c**) The bar charts summarized the calculated numbers of LpM (left) or SpM (middle) and the ratios (SpM/LpM) of these two (right) in each group. Statistical differences between vehicle and drug-treated groups were compared by using *t*-test. The *p* values were presented as *: *p* value < 0.05, and **: *p* value < 0.01.

**Figure 4 biomedicines-10-00185-f004:**
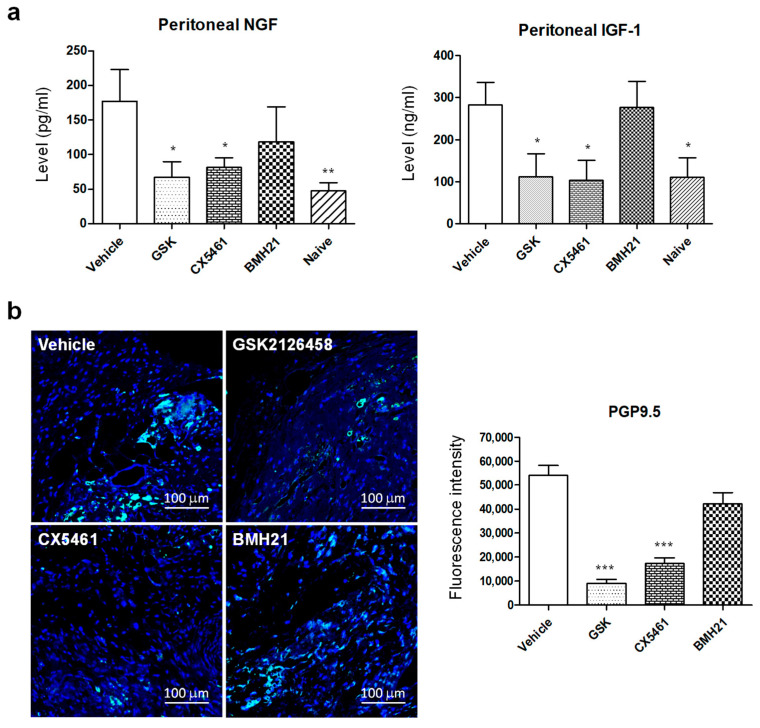
Regulation of nerve fiber growth in endometriotic lesions of mice with induced endometriosis. (**a**) The protein levels of NGF (**left**) and IGF-1 (**right**) in the peritoneal fluid were analyzed by sandwich ELISA. The data were normalized with the average levels in mice treated with vehicle (*n* = 15 for each group). (**b**) Immunofluorescence staining was performed to detect the presence of nerve fibers (PGP9.5^+^) in endometriotic lesions from different groups (**left**). The fluorescence intensity was averaged by using data from 10 independent tissue sections (**right**). Statistical differences between vehicle and drug-treated groups were compared by *t*-test. The *p* values were presented as *: *p* value < 0.05, **: *p* value < 0.01, and ***: *p* value < 0.001.

**Figure 5 biomedicines-10-00185-f005:**
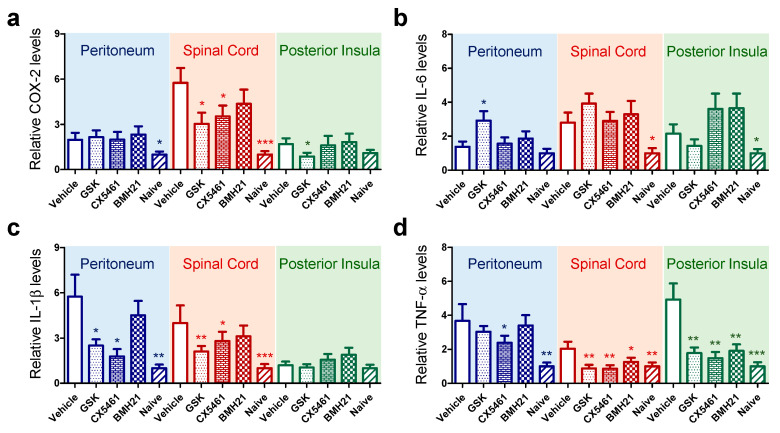
Expression of inflammatory mediators in peritoneal tissue, spinal cord and brain. RNA was extracted from peritoneal tissue (blue), spinal cord (red) and posterior insula (green) in the brain of experimental mice and subjected to cDNA synthesis. qPCR was performed on cDNA samples (*n* = 6 for each group) to detect the expression levels of inflammatory mediators, including (**a**) COX-2, (**b**) IL-6, (**c**) IL-1β and (**d**) TNF-α in different tissues (Appendix A). The data were normalized with the average levels in naïve mice. Statistical differences between vehicle and drug-treated groups were compared by using *t*-test. The *p* values were presented as *: *p* value < 0.05, **: *p* value < 0.01 and ***: *p* value < 0.001.

**Figure 6 biomedicines-10-00185-f006:**
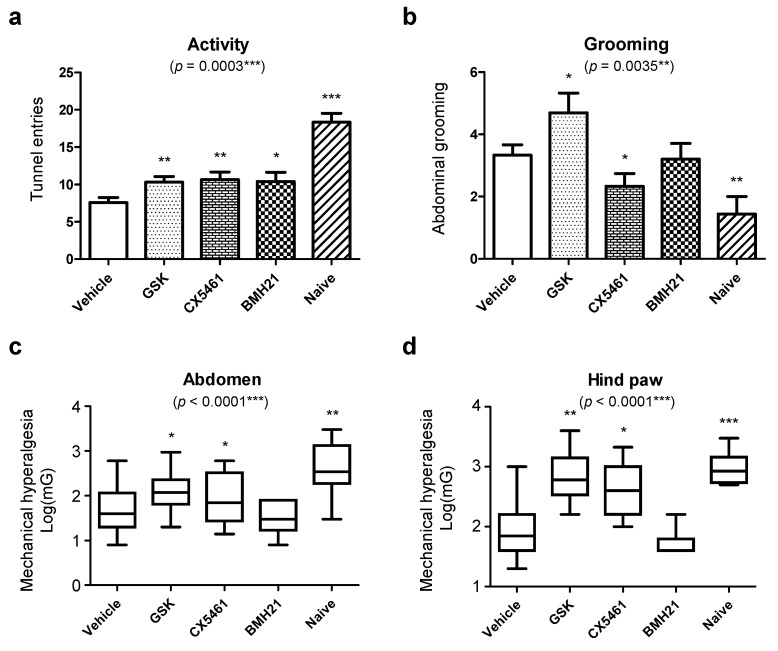
Impacts of anti-ribosome biogenesis agents on pain relief in endometriosis mice. The behavior study was performed to monitor (**a**) tunnel-entering activity and (**b**) abdominal grooming of disease mice treated with different drugs within five minutes. Touch test (von Frey) was also performed on (**c**) abdomen and (**d**) hind paws of mice to quantify pain-sensitivity. The vehicle-treated mice were utilized as the untreated controls whereas naïve mice were utilized as healthy controls. The data were averaged from three independent experiments (*n* = 15 for each group). Statistical differences between vehicle and drug-treated groups were compared by using one-way ANOVA (Kruskal-Wallis) test. The *p* values were presented as *: *p* value < 0.05, **: *p* value < 0.01, and ***: *p* value < 0.001.

**Figure 7 biomedicines-10-00185-f007:**
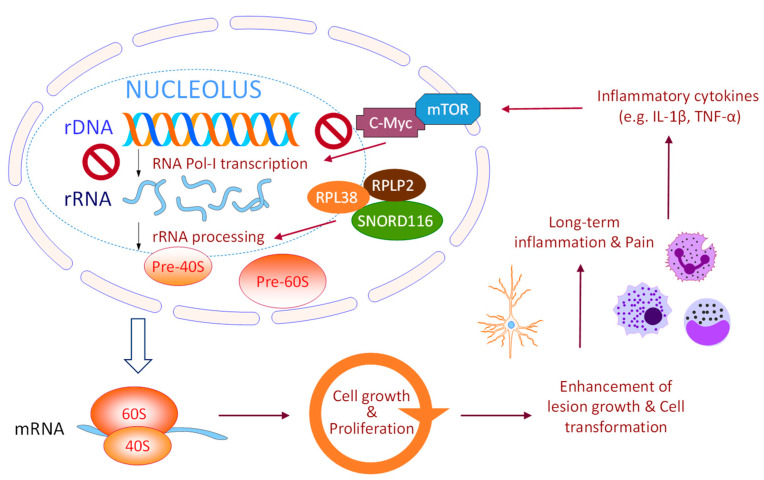
Ribosome biogenesis serves as a therapeutic target for treating endometriosis. The ribosome biogenesis machinery in cells determines energy homeostasis, which is critical to maintain cell growth and proliferation. Endometriosis-induced inflammatory cytokines, e.g., IL-1β and TNF-α, can activate PI3K/AKT/mTOR signaling [34], which subsequently promotes RNA polymerase I-mediated rRNA transcription and editing/processing, resulting in continuous cell growth. Development of endometriotic lesions in peritoneal cavity triggers long-term inflammation and neuron fiber formation, leading to chronic pelvic pain. Up-regulation of ribosome biogenesis by endometriosis-related effectors, such as SNORD116, RPLP2 and RPL38 [12], may speed up the whole process and provide sufficient energy for further aggressive progression. Blockage of ribosome biogenesis by inhibitors against PI3K/mTOR or RNA polymerase-1 can cause genometoxic shock due to energy shortage and ribosomal protein imbalance, leading to cell cycle arrest and apoptosis via p53/p21 pathway.

## Data Availability

Data from the experiments presented in this study are included in this published article and its Appendix A.

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
