# Peer review of "Ribosome Biogenesis Serves as a Therapeutic Target for Treating Endometriosis and the Associated Complications"

_biomedicines, 2022, doi:10.3390/biomedicines10010185_

Round 1

Reviewer 1 Report

The manuscript presented by Chang and collaborators is well written and illustrated. The experimental design and results support the authors' conclusions. Given no therapeutic efficacy of hormone therapy in endometriosis, the inhibition of ribosome biogenesis is important in this context and can create a bridge between pharmacology and therapy.

Author Response

Thanks for the nice words about our study. We highly appreciate your kind attention and support.

Reviewer 2 Report

In my opinion, the article is well written and the experimental design is well elaborated, with very interesting results. This article provides molecular evidence to address whether ribosome biogenesis activation functions as a driving force to promote endometriosis pathogenesis and whether it could be utilized as a newly defined target for treating this debilitating disorder. It reveals the potential of ribosome biogenesis as a potential therapeutic target for treating endometriosis. I have some suggestions for the authors to consider in revising their manuscript:

  • Please include the number of rats used in this study in the materials and methods section.
  • Please provide the name of the ethical approval committee/Institutional Review Board's name and approval number/ID.
  • Please provide a detailed explanation of the treatment model in the materials and methods section
  • Please explain how you analyze the fluorescence staining intensity

Author Response

Thanks for the positive words about our study. We highly appreciate your kind attention and support. Our responses to the comments can be found as the following,

  1. Please include the number of rats used in this study in the materials and methods section.

Response: Power analyses based on our pilot data suggest n=20-25 animals per treatment that provides 80% power to detect 5% significance in this study. We have added this information in section 2.2. “Mouse model of induced endometriosis”, lines 116-117 in the revised manuscript.

  1. Please provide the name of the ethical approval committee/Institutional Review Board's name and approval number/ID.

Response: Thank you for this important consideration. Yes, this information can be found in the Institutional Review Board Statement, lines 418-422 in the revised manuscript.

  1. Please provide a detailed explanation of the treatment model in the materials and methods section

Response: The conditions for drug treatments were designed based on protocols utilized in the previous studies [21,22], and they were 75 µg in 100 µl/mouse/day for GSK2126458 (mTOR/PI3K inhibitor) and 1.25 mg in 100 µl/mouse/day for RNA polymerase-I inhibitors including CX5461 and BMH-21. The drugs were administered orally five days per week for three weeks (Fig. 1a). This information was added in section 2.2. “Mouse model of induced endometriosis”, lines 117-121 in the revised manuscript. Two references were inserted here, thus the refs were renumbered again.

  1. Please explain how you analyze the fluorescence staining intensity?

Response: The fluorescence staining intensity was analyzed by Image-Pro Premier (Media Cybernetics Inc., Rockville, MD). This information was added in section 2.7. “Immunohistochemistry (IHC) & immunofluorescent (IF) staining”, lines 159-160 in the revised manuscript.

Inserted Refs are

  1. Knight, S.D., Adams, N.D., Burgess, J.L., Chaudhari, A.M., Darcy, M.G., Donatelli, C.A., Luengo, J.I., Newlander, K.A., Parrish, C.A., Ridgers, L.H., et al. Discovery of GSK2126458, a Highly Potent Inhibitor of PI3K and the Mammalian Target of Rapamycin. ACS Med Chem Lett 2010, 1, 39-43.
  2. Bywater, M.J., Poortinga, G., Sanij, E., Hein, N., Peck, A., Cullinane, C., Wall, M., Cluse, L., Drygin, D., Anderes, K., et al. Inhibition of RNA polymerase I as a therapeutic strategy to promote cancer-specific activation of p53. Cancer Cell 2012, 22, 51-65.

Reviewer 3 Report

I read with great interest the manuscript, which falls within the aim of this Journal. In my honest opinion, the topic is interesting enough to attract the readers’ attention. Nevertheless, authors should clarify some points and improve the discussion, as suggested below.

Authors should consider the following recommendations:

  • Manuscript should be further revised in order to correct some typos and improve style.
  • Accumulating evidence suggests that immune cells, adhesion molecules, extracellular matrix metalloproteinase and pro-inflammatory cytokines activate/alter peritoneal microenvironment, creating the conditions for differentiation, adhesion, proliferation and survival of ectopic endometrial cells. I would discuss these points in the light of new theories about the pathogenesis of endometriosis, referring to: PMID: 31663401; PMID: 28100109.

Author Response

Thanks for the positive words about our study. We highly appreciate your kind attention and support. Our responses to the comments can be found as the following,

  1. Manuscript should be further revised in order to correct some typos and improve style.

Response: Thank you very much for this suggestion. We have carefully checked the entire manuscript and made some corrections in the revised file.

  1. Accumulating evidence suggests that immune cells, adhesion molecules, extracellular matrix metalloproteinase and pro-inflammatory cytokines activate/alter peritoneal microenvironment, creating the conditions for differentiation, adhesion, proliferation and survival of ectopic endometrial cells. I would discuss these points in the light of new theories about the pathogenesis of endometriosis, referring to: PMID: 31663401; PMID: 28100109.

Response: Thank you for this constructive suggestion. We have added a new paragraph in the Discussion section, lines 385-392, to address new theories about the pathogenesis of endometriosis. Three references [56-58] were inserted here, thus the refs were renumbered again.

Inserted Refs are

  1. Laganà, A.S., Salmeri, F.M., Ban Frangež, H., Ghezzi, F., Vrtačnik-Bokal, E., Granese, R. Evaluation of M1 and M2 macrophages in ovarian endometriomas from women affected by endometriosis at different stages of the disease. Gynecol Endocrinol 2020, 36, 441-4.
  2. Laganà, A.S., Salmeri, F.M., Vitale, S.G., Triolo, O., Götte, M. Stem Cell Trafficking During Endometriosis: May Epigenetics Play a Pivotal Role? Reprod Sci 2018, 25, 978-9.
  3. Pluchino, N., Taylor, H.S. Endometriosis and Stem Cell Trafficking. Reprod Sci 2016, 23, 1616-9.